# Inositol (1,4,5)-Trisphosphate Receptors in Invasive Breast Cancer: A New Prognostic Tool?

**DOI:** 10.3390/ijms23062962

**Published:** 2022-03-09

**Authors:** Arthur Foulon, Pierre Rybarczyk, Nicolas Jonckheere, Eva Brabencova, Henri Sevestre, Halima Ouadid-Ahidouch, Lise Rodat-Despoix

**Affiliations:** 1Centre de Gynécologie-Obstétrique, Université Picardie Jules Verne, CHU Amiens Picardie, F-80089 Amiens, France; 2Laboratoire de Physiologie Cellulaire et Moléculaire, UR-UPJV-4667, Université Picardie Jules Verne, F-80090 Amiens, France; rybarczyk.pierre@chu-amiens.fr (P.R.); henrisevestre@gmail.om (H.S.); halima.ahidouch-ouadid@u-picardie.fr (H.O.-A.); lise.despoix@u-picardie.fr (L.R.-D.); 3Service d’Anatomie et Cytologie Pathologiques, Université Picardie Jules Verne, CHU Amiens Picardie, F-80090 Amiens, France; 4UMR9020-U1277, CANTHER-Cancer Heterogeneity, Plasticity and Resistance to Therapies, CNRS, Inserm, CHU Lille, Univ. Lille, F-59000 Lille, France; nicolas.jonckheere@inserm.fr; 5Jean-Godinot Institute Cancer Center, Centre de Lutte Contre le Cancer de Reims et Champagne Ardenne, Centre de Ressources Biologiques, F-51100 Reims, France; eva.brabencova@reims.unicancer.fr

**Keywords:** inositol 1,4,5 trisphosphate, breast cancer, invasive prognostic marker

## Abstract

**Simple Summary:**

The inositol-trisphosphate receptor (IP_3_R) is a key player in physiological and pathological intracellular calcium signaling. The objective of the present study was to assess the putative value of the three IP_3_R subtypes as prognostic biomarkers in breast cancer. We found that IP_3_R3 is the most strongly expressed subtype in breast cancer tissue. Furthermore, IP_3_R3 and IP_3_R1 are significantly more expressed in invasive breast cancer tissue than in non-tumor tissue. In contrast to IP_3_R1 and IP_3_R2, the expression of IP_3_R3 was positively correlated with prognostic factors including tumor size, regional node invasion, histologic grade, proliferation index, and hormonal status. By analyzing public databases, we found that the expression of all IP_3_R subtypes is significantly correlated with the overall survival and disease-free survival of patients with breast cancer. We conclude that relative to the other two IP_3_R subtypes, IP_3_R3 expression is upregulated in breast cancer and is correlated with prognostic factors. We strongly believe that our results will open up new perspectives with regard to the link between IP_3_Rs and breast cancer aggressiveness.

**Abstract:**

Breast cancer is the leading cause of cancer death among women in worldwide and France. The disease prognosis and treatment differ from one breast cancer subtype to another, and the disease outcome depends on many prognostic factors. Deregulation of ion flux (especially Ca^2+^ flux) is involved in many pathophysiology processes, including carcinogenesis. Inside the cell, the inositol-trisphosphate receptor (IP_3_R) is a major player in the regulation of the Ca^2+^ flux from the endoplasmic reticulum to the cytoplasm. The IP_3_Rs (and particularly the IP_3_R3 subtype) are known to be involved in proliferation, migration, and invasion processes in breast cancer cell lines. The objective of the present study was to evaluate the potential value of IP_3_Rs as prognostic biomarkers in breast cancer. We found that expression levels of IP_3_R3 and IP_3_R1 (but not IP_3_R2) were significantly higher in invasive breast cancer of no special type than in non-tumor tissue from the same patient. However, the IP_3_R3 subtype was expressed more strongly than the IP_3_R1 and IP_3_R2 subtypes. Furthermore, the expression of IP_3_R3 (but not of IP_3_R1 or IP_3_R2) was positively correlated with prognostic factors such as tumor size, regional node invasion, histologic grade, proliferation index, and hormone receptor status. In an analysis of public databases, we found that all IP_3_Rs types are significantly associated with overall survival and progression-free survival in patients with breast cancer. We conclude that relative to the other two IP_3_R subtypes, IP_3_R3 expression is upregulated in breast cancer and is correlated with prognostic factors.

## 1. Introduction

In 2020, around 2.2 million new cases of breast cancer (BC) were diagnosed worldwide [1]—making this disease a major public health problem that affects 1 in 9 women at some point in life. Despite recent and constant progress in diagnosis and management, BC is the second deadliest cancer in women. However, the implementation of BC screening programs and the development of systemic treatments have reduced mortality and the incidence of metastatic cancer. Ninety percent of BC deaths are due to metastases [2,3]. Invasive breast carcinoma of no special type (IBC-NST) is the most frequent histologic subtype. Many prognostic criteria are applied when choosing an adjuvant treatment in BC; these criteria are variously clinical (tumor size, axillary lymph node invasion, remote metastases, etc.) and histologic (the Scarff-Bloom-Richardson (SBR) grade, hormone receptor (HmR) status, human epidermal growth factor 2 (HER2) expression, and the Ki67 index) [4,5,6,7,8,9,10,11]. BC is a complex, heterogeneous disease. Perou et al. developed a molecular classification for BC, which included luminal A, luminal B, HER2, and triple-negative (TN) subtypes [12]. The luminal A and B subtypes generally spread more slowly and recur less frequently than the HER2 and TN subtypes [13,14,15]. Moreover, disease-free survival rates are higher for metastatic luminal BC than for metastatic HER2 or TN BCs [13].

At BC diagnosis, 5% of patients have metastases [16]. Ten to 15% of BC patients will develop remote metastases in the first three years after diagnosis [16,17]. A third of patients without axillary lymph node involvement will still develop metastases [17]. Early-stage (non-metastatic) BC is curable; treatments include local surgery, brachytherapy, and systemic therapies such as chemotherapy and hormone therapy. The choice of treatment for BC is now driven by the tumor’s histologic and molecular characteristics. The current policy of care for BC is tending towards less aggressive, more targeted, more personalized treatments [18]. Indeed, the indication for systemic chemotherapy is based on the risk of recurrence risk. This risk is determined by the above-mentioned risk factors and the BC’s molecular classification. This therapeutic de-escalation has been made possible by the development of new tools for determining the indication for adjuvant chemotherapy. Genomic tests can now evaluate the 10-year risk of BC recurrence, which in turn can guide the treatment choice [18].

Despite the development of these new tools, some BCs will recur locally or will form metastases. It is therefore essential to find novel treatments and prognostic markers for even more effective patient care. The heterogeneity of BC is now taken into account when developing new therapies and prognostic factors, with a view to making treatment even more personalized.

Calcium ions (Ca^2+^, from outside the cell or from within the endoplasmic reticulum (ER)) drive the development of metastases. When Ca^2+^ channels are open, the cytosolic Ca^2+^ concentration rises by a factor of 5 to 10 (from 100 nM to 500–1000 nM). Inside the cell, inositol (1,4,5)-trisphosphate (IP_3_) generates part of this calcium signaling via the IP_3_ receptor (IP_3_R), which has three characterized subtypes: IP_3_R1, IP_3_R2, and IP_3_R3. The three subtypes show 60% to 80% homology and are expressed ubiquitously. Interestingly, it was found that each subtype has a specific calcium release signature: strong Ca^2+^ oscillations through IP_3_R2, weaker oscillations through IP_3_R1, and monophasic transients through IP_3_R3 [19].

The (dys)regulation of IP_3_Rs expression and activity is involved in many oncogenic processes, including cancer cell growth, migration, proliferation, and survival. IP_3_R1 is involved in apoptosis resistance in prostate cancer cells [20]. IP_3_R2 is overexpressed in acute myeloid leukemia, and the level of expression is significantly correlated with poorer overall survival [21]. IP_3_R3 is involved in glioblastoma cell migration and invasion [22], gastric cancer cell proliferation [23], and many other cancers (pancreatic, colonic, and renal) [24]. 

With regard to BC more specifically, the IP_3_R3 overexpression induced by estradiol promotes MCF-7 BC cell growth in vitro [25]. IP_3_R3 also regulates BC cell line proliferation via an interaction with BKCa voltage- and Ca^2+^-dependent K^+^ channels [26]. Moreover, IP_3_R3 expression increases the migration capacity of human BC cells by shifting calcium oscillations towards a more sustained signature [27]. IP_3_R3 is also able to coordinate the remodeling of the profilin cytoskeleton organization through the ARHGAP18/RhoA/mDia1/FAK pathway [28]. In human BC more specifically, IP_3_R1 is not overexpressed. In contrast, Singh et al. found that IP_3_R2 and IP_3_R3 are more highly expressed in BC tissue than in non-tumor tissue [29]. In this context, we sought to (i) characterize the expression patterns of all IP_3_R subtypes within human BC tissue and (ii) evaluate the putative correlation between the IP_3_R3 expression level and the BC’s proliferative/aggressive profile.

## 2. Materials and Methods

We conducted a prospective, observational study (named CARCINO study) at Amiens University Medical Center (Amiens, France). The study was approved by the local institutional review board (CPP Nord-Ouest II, Amiens, France; reference: ID-RCB 2015-A00537-42, dated July 2015). We included patients requiring surgery for an invasive BC with a greatest dimension > 15 mm. After resection, the pathologist collected one to three tumor tissue samples (size: 3 to 7 mm) depending on the tumor size. At least one sample per patient included was frozen immediately and stored at −80 °C for Western blot assays, any other samples were then used for immunohistochemistry. Informed consent was obtained from all subjects involved in the study.

### 2.1. Western Blot

The frozen tissue sample was mechanically dissociated in RIPA buffer (1% Triton 100×, 1% sodium deoxycholate, 150 mM NaCl, 2 mM EDTA, 5 mM PO_4_Na_2_/K, pH 7.2) supplemented with 0.8% protease inhibitor cocktail (Sigma Aldrich, Il, USA) in a special test tube for dissociation, using the GentleMACS^TM^ system (Miltenyi Biotec, MA, USA) and the “Protein 01_01” protocol. After centrifugation at 15,000× *g* and 4 °C for 15 min, the protein concentration in the supernatant was assayed using the BCA method (Bio-Rad, CA, USA) according to the manufacturer’s instructions. Protein samples were then denatured for 10 min at 95 °C in Laemmli sample buffer. Protein was separated by SDS-PAGE and transferred onto nitrocellulose membranes (Hybond, Wis, GE Healthcare, Chicago, IL, USA). Membranes were blocked in 1% BSA in TBS-T (0.1% Tween 20, 50 mM Tris HCl buffer, 150 mM NaCl, pH 7.5). Next, the membranes were incubated overnight at 4 °C with mouse monoclonal anti-IP_3_R1 (1/500, Neuromab, CA, USA), rabbit monoclonal anti-IP_3_R2 (1/250, Santa Cruz, CA, USA), mouse monoclonal anti-IP_3_R3 (1/500, BD Biosciences, Switzerland), or goat polyclonal anti-actin (1/2500, Santa Cruz, CA, USA) primary antibodies diluted in 1% BSA in PBS-T. Actin primary antibody was used for loading control experiments. Membranes were then incubated for 1 h at RT with respective secondary antibodies (1/2500–1/5000; Santa-Cruz, CA, USA), developed using ECL substrate solution (ECL, RevelBolt Intense, Cell Signaling, Neve Yamin, Israel), exposed with the MF-ChemiBIS (DNR, bio-imaging systems, Neve Yamin, Israel) and analyzed using Quantity One software (Biorad, Hercules, CA, USA). 

### 2.2. Immunohistochemistry

Formalin-fixed, paraffin-embedded sections of BC tissue (thickness: 2 to 3 µm) were deparaffinized in xylene and then rehydrated in ethanol. The endogenous peroxidase activity was blocked before the antigen retrieval. The cell conditioning solution CC1 (BenchMark XT, Ventana, Rotkreuz, Switzerland) was then used for antigen retrieval.

Immunohistochemical staining was carried out on a BenchMark ULTRA system (Ventana, Rotkreuz, Switzerland) using antibodies against the three IP_3_Rs (1/50 for IP_3_R1 (Neuromab, CA, USA), 1/50 for IP_3_R2 (Santa Cruz, CA, USA) and 1/100 for IP_3_R3 (BD Biosciences, Switzerland)). This was followed by avidin–biotin–peroxidase complex treatment. The signals were developed using a chromogenic reaction with 3,3′-diaminobenzidine tetrahydrochloride (iVIEW DAB Detection Kit, Ventana). The tissues were counterstained with hematoxylin. All antibodies were certified for immunohistochemical use. All experiments included a negative control (without the primary antibody).

The results were rated independently by two experienced investigators (PR and AF), using a Leica inverted microscope. The staining intensity score ranged from 0 to 3 (0 = no immunostaining; 1 = weak immunostaining; 2 = moderate immunostaining; 3 = strong immunostaining), and the percentage of stained cells was also recorded. An IP_3_R immunohistochemical (IH) expression score was then attributed for each tissue sample by multiplying the intensity score by the percentage of stained cells. The IH expression score therefore ranged from 0 (lowest) to 3 (highest).

We also used immunohistochemistry to assess the molecular subtype. This classification is essentially based on positivity for (and the percentage expression of) HmRs (the ER, in particular), HER2, and Ki67 (Table 1). 

### 2.3. Survival Analysis

Survival analysis was conducted using the SurvExpress online tool (Available in bioinformatica.mty.itesm.mx/SurvExpress). Expression levels of the individual genes (*ITPR1*, *ITPR2*, and *ITPR3*) and the combined signature were analyzed using SurvExpress and the optimized Maximize algorithm, which attributes a minimum *p*-value to a risk group. The hazard ratio (HR) [95% confidence interval (CI)] was also evaluated. Five datasets were used: the “Breast cancer recurrence data, 9 datasets from 7 authors” (1561 patients), “*Breast cancer Meta-base: 10 cohorts 22K gene*” (1888 patients), “*Breast Invasive Carcinoma TCGA”* (502 patients), “*Miller Bergh Breast GSE3494-GPL96”* (236 patients) and “*BRCA-TCGA Breast invasive carcinoma*—*July 2016”* (962 patients).

### 2.4. Statistical Analyses

In a descriptive analysis, normally distributed quantitative variables were quoted as the mean ± standard error of the mean (SEM). Pairs of mean values were compared using a non-parametric Mann–Whitney test. A non-parametric Kruskal–Wallis test was used to compare means of more than two groups. The threshold for statistical significance was set to *p* < 0.05.

## 3. Results

### 3.1. Study Population

Between 1 November 2015 and 1 November 2018, 52 patients with IBC-NST treated at Amiens University Medical Center were included in the study. An additional 15 IBC-NST samples from patients at the Jean Godinot Institute cancer center (Reims, France) were included. Non-tumor tissue from the same patient was available for each BC tissue sample. The clinical and histologic characteristics are summarized in Table 2.

### 3.2. IP_3_Rs and Invasive Breast Carcinoma of No Special Type

First, we used Western blotting to evaluate the expression of each IP_3_R subtype in IBC-NST samples and non-tumor tissue samples. We found that both IP_3_R1 and IP_3_R3 expression were significantly higher in BC tissue than in non-tumor tissue (IP_3_R1; 1.59 ± 0.04 (*N =* 26) vs. 1 ± 0.03 (*N =* 12), respectively; *p* = 0.02; IP_3_R3: 3.37 ± 0.14 (*N =* 29) vs. 1 ± 0.02 (*N =* 12), respectively; *p <* 0.0001) (Figure 1A). IP_3_R3 expression was three times greater in BC tissue than in non-tumor tissue. In contrast, there was no difference in IP_3_R2 expression between BC tissue and non-tumor tissue (1.05 ± 0.06 (*N =* 25) vs. 1 ± 0.08 (*N =* 13), respectively; *p* = 0.8) (Figure 1A). The same results were obtained when considering the IH expression score. Indeed, the IP_3_R1 IH expression score was significantly higher in BC tissue than in non-tumor tissue (0.75 ± 0.14 (*N =* 40) vs. 0.39 ± 0.14 (*N =* 18), respectively; *p* = 0.05); the same was true for the IP_3_R3 IH expression score (1.48 ± 0.12 (*N =* 41) vs. 0.22 ± 0.08 (*N =* 17), respectively; *p* < 0.0001) (Figure 1B). In contrast, there was no difference in the IP_3_R2 IH expression score between BC tissue and non-tumor tissue (1.64 ± 0.11 (*N =* 30) and 1.43 ± 0.19 (*N =* 18), respectively; *p* = 0.3) (Figure 1B). Taken together, our results showed that IP_3_R1 and IP_3_R3 are overexpressed in IBC-NST, but not IP_3_R2.

### 3.3. IP_3_R Expression and Predictive Factors

After having found, in Western blot and in IH, that IP_3_R1 and IP_3_R3 were overexpressed in IBC-NST, we next sought to establish a link between IP_3_R expression on the one hand and predictive factors for survival and recurrence in BC patients on the other.

### 3.4. IP_3_R Subtype IH Expression and Tumor Size

Tumor size is a major risk factor for local or remote BC recurrence; the larger the tumor, the greater the risk [30,31,32,33]. We compared the IH expression score for the three subtypes, as a function of the tumor size. We also studied the IH expression of IP_3_R2, even though its expression levels were similar in the paired tumor and non-tumor samples.

In BCs less than 20 mm in size (T1), there were no statistically significant differences in the IP_3_R1, IP_3_R2, and IP_3_R3 IH expression scores (IH expression score in T1 BCs: 1.08 ± 0.22 (*N* = 10) for IP_3_R1; 1.67 ± 0.21 (*N* = 9) for IP_3_R2 and 1.36 ± 0.3; *p* = 0.33 (*N* = 10) for IP_3_R3) (Figure 2A). In T2 BCs (20–50 mm) and T3 BCs (>50 mm), the IP_3_R2 and IP_3_R3 IH expression scores were significantly higher than the IP_3_R1 IH expression score (IH expression score in T2 BCs: 0.73 ± 0.18 (*N* = 24) for IP_3_R1 vs. 1.63 ± 0.14 (*N* = 19); *p* = 0.0001 for IP_3_R2 and 1.4 ± 0.14 (*N* = 25); *p* = 0.004 for IP_3_R3. IH expression score in T3 BCs: 0.18 ± 0.12 (*N* = 5) for IP_3_R1 vs. 1.8 ± 0.2 (*N* = 3); *p* = 0.04 for IP_3_R2 and 1.92 ± 0.32 (*N* = 5); *p* = 0.008 for IP_3_R3) (Figure 2B,C). The IH expression of IP_3_R2 was independent of tumor size. Thus, large tumor size was more closely related to IP_3_R3 expression than to IP_3_R1 expression.

### 3.5. IP_3_Rs and Lymph Node Involvement

Regional lymph node involvement is a risk factor for local and remote recurrence [30,31]. We compared IBC-NST with (N+) and without (N0) regional lymph node involvement with regard to the IH expression score for each of the three subtypes.

Only the IP_3_R3 IH expression score was greater (by a factor of 1.3) in N+ BCs (1.72 ± 0.16 (N = 20)) than in N0 BCs (1.3 ± 0.16 (N = 21); *p* = 0.08) (Figure 3A). The IP_3_R1 and IP_3_R2 IH expression scores were similar when comparing N0 and N+ samples. In N0 BCs, the IH expression score was 0.77 ± 0.14 (*N =* 20) for IP_3_R1 vs. 1.78 ± 0.17 (*N =* 16; *p <* 0.0001) for IP_3_R2, and 1.3 ± 0.16 (*N =* 21; *p* = 0.03) for IP_3_R3. In N+ BCs, the IH expression score was 0.73 ± 0.21 (*N =* 20) for IP_3_R1, vs. 1.49 ± 0.13 (*N =* 14; *p* = 0.001) for IP_3_R2, and 1.72 ± 0.16 (*N =* 20; *p* = 0.001) for IP_3_R3. (Figure 3B). Moreover, the IP_3_R1 IH expression score was significantly lower than the IP_3_R2 and IP_3_R3 IH expression scores in both N0 and N+ BCs (Figure 3B). Thus, IP_3_R3 appeared to be specifically related to lymph node involvement in IBC-NST. 

### 3.6. IP_3_Rs, Histology Grades, and the Ki67 Proliferation Index

The Ki67 proliferation index and the histologic (SBR) grade are risk factors for BC recurrence: the higher the index or grade, the greater the risk of recurrence [5,30,31,32,33]. We therefore compared the SBR histologic grade and Ki67 index with regard to the three subtypes IH expression scores. We found that the IH expression scores for IP3R2 and IP3R3 were significantly higher than that for IP3R1 in grade III SBR samples and samples with a Ki67 index greater than 20% (Figure 4A,B). The IH expression score in grade 3 samples was 0.46 ± 0.12 (*N* = 16) for IP3R1 vs. 1.47 ± 0.26 (*N* = 9; *p* = 0.0006) for IP3R2 and 1.64 ± 0.19 (*N* = 17; *p* < 0.0001) for IP3R3. (B) The IH expression score in samples with a Ki67 index > 20% was 0.59 ± 0.2 (*N* = 20) for IP3R1 vs. 1.75 ± 0.15 (*N* = 13; *p* < 0.0001) for IP3R2 and 1.59 ± 0.18 (*N* = 21; *p* < 0.0001) for IP3R3 (Figure 4A,B). In both settings, there was no difference between the IP3R2 and IP3R3 IH expression scores. Given that IP3R2 is not overexpressed in BC tissue, IP3R3 expression is thus closely related to BC with a poor prognosis. 

### 3.7. IP_3_Rs and the Molecular Classification of BCs

At present, clinicians typically classify carcinomas into five subtypes (luminal A, luminal B HER2^−^, luminal B HER2^+^, HER2, and TN) on the basis of histologic and molecular characteristics, which notably include expression of the estrogen receptor (ER), progesterone receptor (PR), HER2, and Ki67. Tumors that are ER^+^ and/or PR^+^ are referred to as HmR^+^. The prognosis, disease aggressiveness, and risk of remote metastasis and relapse vary from one subtype to another. In fact, the luminal A subtype has the best prognosis [34].

We assessed the three IP_3_R IH expression scores in the five BC molecular subtypes. In luminal A BC, there were no significant differences in the IH expression scores for the three IP_3_R subtypes (IH expression scores in luminal A BC: 0.97 ± 0.17 (*N =* 17) for IP_3_R1, vs. 1.61 ± 0.16 (*N =* 14) for IP_3_R2 and: 1.33 ± 0.17 (*N =* 17) for IP_3_R1; *p* = 0.2) (Figure 5A). In luminal B and TN samples, the IH expression score was significantly lower for IP_3_R1 than for IP_3_R2 and IP_3_R3 (IH expression scores in luminal B BC: 0.62 ± 0.21 (*N =* 14) for IP_3_R1, vs. 1.54 ± 0.18 (*N =* 9; *p* = 0.002) for IP_3_R2 and 1.54 ± 0.18 (*N =* 15; *p* = 0.0002) for IP_3_R3 and IP_3_R1 IH expression scores in TN BC: 0.51 ± 0.36 (*N =* 8) for IP_3_R1 vs. 1.97 ± 0.26 (*N =* 6; *p* = 0.01) for IP_3_R2 and 1.48 ± 0.32 (*N =* 8; *p* = 0.02) for IP_3_R3) (Figure 5B,C). In luminal B HER2- samples, the IH expression score was significantly higher for IP_3_R2 than for IP_3_R1 (IH expression scores in luminal B HER2-: 0.66 ± 0.42 (*N =* 7) for IP_3_R1, vs. 1.67 ± 0.17 (*N* = 6; *p* = 0.04) for IP_3_R2 and 1.3 ± 0.31 (*N* = 7) for IP_3_R3). In luminal B HER2+ samples, the IH expression score was significantly higher for IP_3_R3 than for IP_3_R1 (IP_3_R1 IH expression scores in luminal B HER2+: 0.56 ± 0.15 (*N =* 5) for IP_3_R1, vs. 1.77 ± 0.23 (*N =* 6); *p* = 0.004) for IP_3_R3 and 1.38 ± 0.38 (*N* = 4) for IP_3_R2) (Figure 5D,E). There were no differences between the IP_3_R2 and IP_3_R3 IH expression scores in any of the subtypes. This finding suggests that IP_3_R3 is associated with a more aggressive disease profile for IBC-NST. 

### 3.8. Expression of IP_3_Rs in Non-Tumor Tissue

We also compared the expression of IP_3_Rs in the non-tumor tissue. Our results, obtained both by IH and Western blot, show that IP_3_R2 is strongly expressed in contrast to IP_3_R1 and IP_3_R3. IH expression score: IP_3_R2 = 1.43 ± 0.19 (*N* = 18) vs. 0.39 ± 0.14 (*N* = 18); *p* = 0.0001 for IP_3_R1 and 0.22 ± 0.08 (*N* = 17); *p* < 0.0001 for IP_3_R3. The IP_3_R2 relative expression score is 0.65 ± 0.08 (*N* = 13) vs. 0.24 ± 0.03 (*N* = 12); *p* < 0.0001 for IP_3_R1 and 0.25 ± 0.03 (*N* = 12); *p* < 0.0001) for IP_3_R3. Indeed, the relative expression and IH expression scores are significantly higher for IP_3_R2 than for IP_3_R1 or IP_3_R3 (Figure 6A,B). Thus, in contrast to IP_3_R3, IP_3_R2 appears as a highly expressed IP_3_R in non-tumoral as in tumoral breast tissue. 

### 3.9. IP_3_Rs and Patient Survival

We next sought to establish whether or not the expression of the three IP_3_R subtypes was correlated with the patients’ survival. We compared survival data and the HR for predefined populations, such as high-risk and low-risk groups for the three receptor subtypes (Figure 7). We used the SurvExpress website’s optimized algorithm to generate risk groups by sorting according to the prognostic index (a higher IP_3_R value for a higher risk) and splitting the patients into cohorts where the *p*-value was the lowest [35].

We chose to analyze five databases (listed in the “Survival analysis” section of the Methods) with the largest number of documented patients. In the “*BRCA-TCGA Breast invasive carcinoma*—*July 2016*” database, IP_3_R1 and IP_3_R3 expression were significantly associated with poor overall survival (Figure 7A,B). The results for IP_3_R2 were not statistically significant (Figure 7C). IP_3_R1 gave a higher HR (1.8; 95%CI [1.27–2.54]; *p* = 0.0009) than IP_3_R3 (1.67; 95%CI [1.19–2.35]; *p* = 0.003) or IP_3_R2 (1.22; 95%CI [0.85–1.75]; *p* = 0.286) (Figure 7D). IP_3_R1 and IP_3_R3 expression were significantly associated with poor overall survival in 1358 and 1700 patients, respectively (Table 3), whereas IP_3_R2 expression was associated with poor overall survival in only 434 patients (Table 3).

Whenever possible, we evaluated the association between IP_3_RS expression and disease-free survival (Table 3); data were available for the “*Breast cancer recurrence data, 9 datasets from 7 authors*” and “*Breast cancer Meta-base: 10 cohorts 22K gene*” databases, comprising a total of 3449 samples. IP_3_R expression was significantly associated with poor disease-free survival (Table 3).

## 4. Discussion

Our present results evidenced different IP_3_R subtype expression profiles in BC. IP_3_R3 was most overexpressed in IBC-NST and was correlated with clinical features such as tumor size and grade, regional node invasion, proliferation index, and HmR status. IP_3_R1 was also overexpressed, albeit to a lesser extent than IP_3_R3; however, the expression of IP_3_R1 was inversely correlated with tumor size and did not vary as a function of the other prognostic markers. IP_3_R2 was more strongly expressed than the other two subtypes in non-tumor tissue without being overexpressed in BC tissue compared to non-tumor tissue. Interestingly, IP_3_R2 expression remained high regardless of tumor size, lymph node involvement, histologic grade, tKi67 index, and BC molecular classification. Moreover, we found that high IP_3_R expression was significantly associated with worse overall survival among treated patients.

IP_3_Rs have a role in the pathophysiology of cancer. In myeloid acute leukemia, the IP_3_R2 expression level is correlated with worse overall survival [21]. IP_3_R1 is involved in resistance to apoptosis in prostate cancer cells [20] and in the epithelial–mesenchymal transition induced by epidermal growth factor in the MDA-MB-468 human BC cell line [36]. Kang et al. demonstrated that IP_3_R3 had prognostic value in glioblastoma in an animal model; the survival rate was higher when IP_3_R3 expression was inhibited [22]. IP_3_R3 is also involved in the peritoneal dissemination of gastric cancer cells [23]. Moreover, the IP_3_R3 expression level is correlated with the aggressiveness of colorectal cancer [37].

More specifically, IP_3_R3 is a key factor in many mechanisms of BC oncogenesis. It is involved in BC cell proliferation via estrogen-dependent stimuli [25] and an interaction with the BKCa potassium channel [26]. Moreover, the IP_3_R3 expression level influences the migration capacity of human BC cells by changing the calcium signature [27]. IP_3_R3 is also able to coordinate the remodeling of the profilin cytoskeleton via the ARHGAP18/RhoA/mDia1/FAK pathway [28]. However, a prognostic role for IP_3_R3 (or even a correlation with BC aggressiveness, as observed for glioblastoma or colorectal cancer) has not yet been demonstrated. Studying the tissue expression of the three IP_3_R subtypes is thus essential for understanding their mechanism of action. In that context, our results established for the first time a link between the level of expression of IP_3_R3 and the aggressiveness of IBC-NST. To the best of our knowledge, our study is the first to have established this link in BC—a link that is already known for cancers such as glioblastoma and gastric cancer. Although the prospective nature of our study consequently limited the number of samples eligible for this work, it enabled us to obtain precise data in terms of overall survival but also in terms of recurrence-free survival at 5 years. These data will be the subject of a future study.

Based on a SurvExpress analysis of five databases, we showed that expression of the three IP_3_R subtypes is negatively correlated with overall survival and disease-free survival. In BC, there are many clinical or histologic survival factors. All these criteria should be taken into account when selecting the primary treatment, neo-adjuvant or adjuvant chemotherapy, or a more targeted type of therapy such as hormone therapy. The negative correlation between IP_3_R subtype expression on the one hand, and overall survival and disease-free survival on the other, testifies to the receptor’s significant involvement in breast carcinogenesis in general and metastatic invasion in particular. It is known that the mortality rate in BC is linked to the occurrence of remote metastases [38,39]. Moreover, our database analyses showed that receptor expression is inversely correlated with overall survival and disease-free survival—suggesting a genuine prognostic role for these receptors.

We found that IP_3_R1 and (especially) IP_3_R3 are expressed significantly more in BC tissue than in non-tumor tissue (Figure 1). IP_3_R3 expression appears to be correlated with all the known prognostic factors for BC. In non-tumor tissue, IP_3_R2 is expressed significantly more than IP_3_R1 and IP_3_R3; however, IP_3_R2 expression is also high in all BC subtypes. IP_3_R1 is overexpressed in BC but to a lesser extent than IP_3_R3. IP_3_R1 expression (relative to the two other subtypes) decreases with tumor size but is not correlated with other BC aggressiveness factors (lymph node involvement, histologic grade, and the Ki67 index). IP_3_R3 can thus be considered as a marker of aggressiveness in BC as its expression is correlated to the severity of BC.

Our data argue in favor of the involvement of all IP_3_R subtypes in breast carcinogenesis processes. Differences in expression between non-tumor tissue and tumor tissue are also observed in colorectal cancer. Indeed, IP_3_R3 is not detected in non-tumor tissue (in contrast to the other two subtypes) but is overexpressed in colorectal cancer tissue [37]. The IP_3_R subtypes are not expressed in all tissues and are expressed to differing degrees in given cell types. In rat hepatocytes, for example, IP_3_R3 is not expressed, IP_3_R1 is expressed diffusely in the cytoplasm, and IP_3_R2 is concentrated in the pericanalicular region [38]. These subcellular differences in location suggest that the three IP_3_R subtypes have distinct functions in the hepatocyte’s calcium signaling activity [38]. The differences in expression of the three subtypes between BC tissue and non-tumor tissue suggest that each subtype has a different role in breast carcinogenesis. IP_3_R1 and IP_3_R3 are involved in the epithelial–mesenchymal transition in BC cells, whereas IP_3_R2 is not [36].

Next, we sought to establish a link between IP_3_R expression and other predictive factors of survival in BC. There were no significant differences in IP_3_R subtype expression in tissue from BCs smaller than 20 mm. However, the IH expression scores were significantly higher for IP_3_R2 and IP_3_R3 than for IP_3_R1 in tumors larger than 20 mm (i.e., T2 and T3). IP_3_R3 expression appears to be correlated with tumor size. The IP_3_R3 subtype is therefore linked to disease aggressiveness with regard to tumor size—a major factor in the decision to prescribe adjuvant treatment. Based on other prognostic criteria, patients with a small tumor size might not receive chemotherapy. Wallgren et al. showed that tumor size >20 mm was a risk factor for local or remote recurrence [4].

Regional lymph node invasion is a major risk factor for recurrence; the greater the number of invaded axillary lymph nodes, the greater the incidence of local or remote recurrence [4]. IP_3_R3 is the most highly involved subtype since it (but not IP_3_R1 or IP_3_R2) is upregulated in N+ tissue. IP_3_R3 expression may therefore be linked to a more aggressive profile with regard to the lymph node invasion criterion.

In SBR grade III IBC-NST, IP_3_R1 was expressed significantly less than the other two subtypes. These results suggest that IP_3_R3 expression is associated with the histologic prognosis and, again, more aggressive disease in patients with IBC-NST. Likewise, in tissue with a Ki67 proliferation index above 20%, IP_3_R1 was expressed significantly less than the other two subtypes. IP_3_R3, therefore, appears to be involved in IBC-NST with a high proliferation index and is linked to a more aggressive disease profile. The proliferation index and the histologic grade are predictive of a poor prognosis and are taken into account for certain treatment decisions.

Lastly, IP_3_R3 was expressed significantly more than IP_3_R1 in BCs overexpressing HmRs and HER2, whereas the expression of IP_3_R2 was significantly higher than that of IP_3_R1 in TN BC. Moreover, no differences between IP_3_R2 expression and IP_3_R3 expression were found in these two tumor subtypes. Overexpression of the HER2 receptor and the absence of HmR and HER2 expression are associated with a worse prognosis [10]. Our results therefore suggest that IP_3_R3 is associated with more aggressive IBC-NST profiles, as defined by positivity or negativity for HmRs and HER2.

Based on our results, IP_3_R3 expression appears to be associated with more aggressive IBC-NST profiles. We established that the level of IP_3_R3 expression was significantly higher in BC tissue than in non-tumor tissue. IP_3_R2 was not overexpressed in BC tissue but is strongly expressed in all tissues—suggesting that it is essential for the mammary gland’s function. Therefore, our findings suggest that (i) IP_3_R3 is the subtype most significantly involved in breast carcinogenesis processes, and (ii) IP_3_R3 expression is linked to a more aggressive BC profile. Hence, IP_3_R3 might be a specific marker of BC aggressiveness. In contrast to the other two subtypes, IP_3_R3 is significantly involved in the proliferation and migration of human BC cells [26,27]. Furthermore, IP_3_R3 influences the morphology of BC cells. When IP_3_R3 expression is low, human BC cells take on a round shape that is much less conducive to migration and invasion [28]. Proliferation, migration, and invasion are three major factors in carcinogenesis. In BC, the prognosis is essentially linked to the presence of regional or remote metastases. When cancer cells lack migratory or invasive abilities, they are unable to metastasize. IP_3_R3 might be involved in cell migration and invasion, with a role in the BC prognosis. IP_3_R3 is known to be involved in the carcinogenesis of other cancers. IP_3_R3 is thus overexpressed in cholangiocarcinoma, where it is involved in cell migration and proliferation [39]. It also has a major role in invasion processes in glioblastoma [22]. These two cancers have a very poor prognosis, which emphasizes IP_3_R3 putative involvement as a prognostic factor in certain cancers.

IP_3_Rs (mainly IP_3_R3) have a role in apoptosis. In B and T lymphocytes, elevated IP_3_R3 expression induces more apoptosis [40]; this is also the case in pancreatic cells [41] and hepatocytes [42]. In some cancers, IP_3_R3 becomes anti-apoptotic; this is the case in clear cell kidney cancers, where IP_3_R3 has anti-apoptotic activity and IP_3_R1 has pro-apoptotic activity [43]. This might also be the case in BC since the inhibition of IP_3_R expression in human BC cells induces an increase in apoptosis [29].

The current problem in BC management is the absolute necessity to not “over-treat” patients. The latest results from basic research and clinical studies argue in favor of therapeutic de-escalation and personalized medicine [18]. For example, the advent of genomic testing several years ago has enabled some patients with non-aggressive cancers to avoid chemotherapy. IP_3_R3′s involvement in overall survival and disease-free survival in BC remains to be defined. If it is proven that a low IP_3_R3 threshold in an invasive breast tumor is correlated with greater survival, this receptor might eventually become a complementary tool in the decision to initiate (or not) adjuvant treatment. IP_3_R3 expression could therefore be integrated into genomic-based calculations of the recurrence risk.

In conclusion, our results strongly suggest that IP_3_R3 can be considered a prognostic marker in BC. In the longer term, it might be possible to predict tumor aggressiveness and perhaps select an appropriate adjuvant treatment if the primary tumor strongly expresses IP_3_R3.

## Figures and Tables

**Figure 1 ijms-23-02962-f001:**
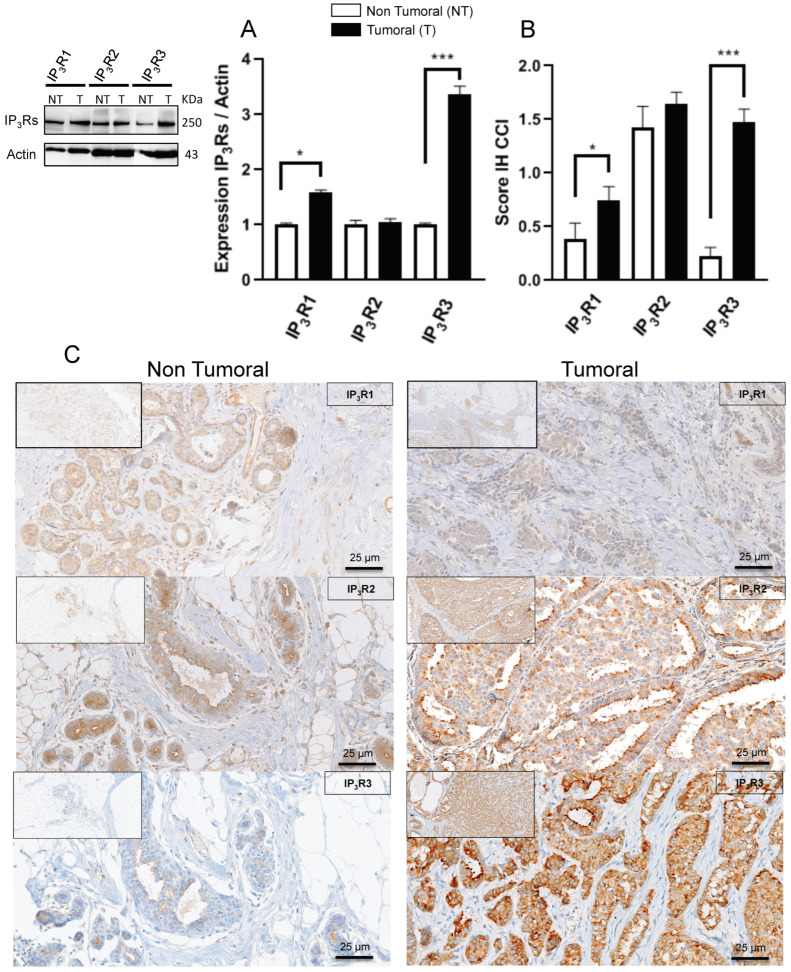
IP_3_R expression in BC tissue and non-tumor tissue. The relative expression levels of IP_3_R1 and IP_3_R3 are significantly higher in BC tissue than in non-tumor tissue; this difference was not observed for IP_3_R2 (**A**). The same results were obtained when considering the IH expression score (**B**). (**A**) IP_3_R relative expression in IBC-NST, in a Western blot (T: tumor tissue; NT: non-tumor tissue). (**B**) The IP_3_R IH expression score in IBC-NST (T: tumor tissue; NT: non-tumor tissue). (**C**) A representative IH image (magnification: 200 X; insert: 800 X). * *p* < 0.05; *** *p* < 0.001.

**Figure 2 ijms-23-02962-f002:**
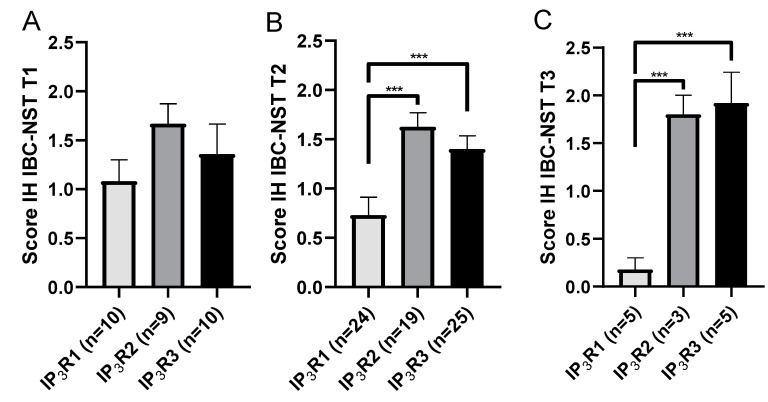
The IP_3_R IH expression score, as a function of tumor size (**A**–**C**). The IP_3_R IH expression scores did not differ significantly in T1 BC (**A**). The IP_3_R2 and IP_3_R3 IH expression scores were significantly higher than IP_3_R1 score in T2 and T3 BCs (**B**,**C**). *** *p* < 0.001.

**Figure 3 ijms-23-02962-f003:**
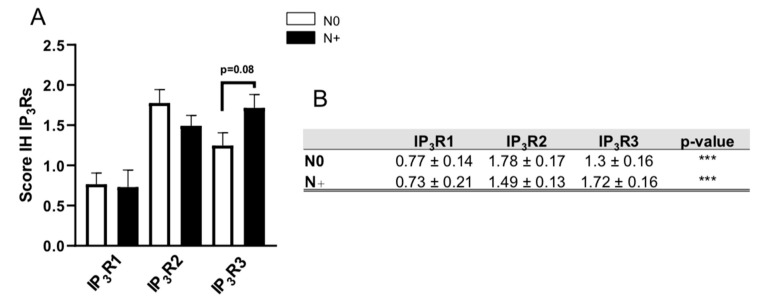
The IP_3_R IH expression scores as a function of lymph node status (**A**,**B**). *** *p* < 0.001.

**Figure 4 ijms-23-02962-f004:**
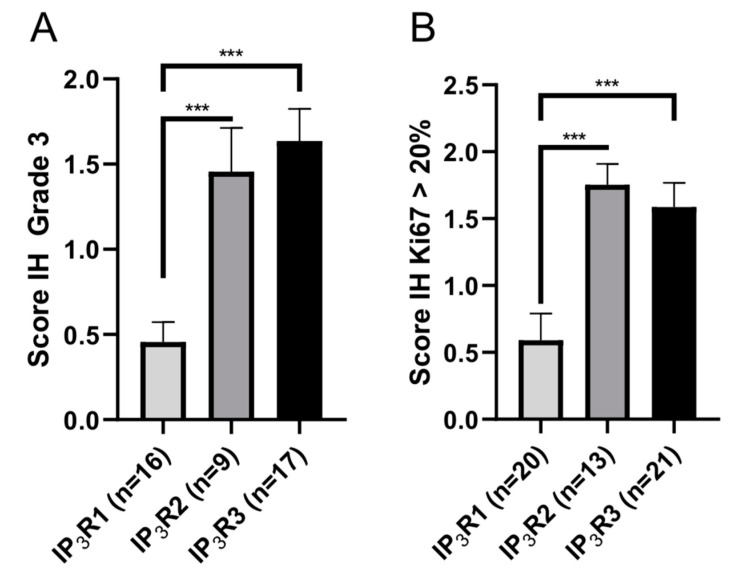
The IP3R IH expression scores in SBR grade 3 IBC-NST (**A**) and IBC-NST with a high Ki67 proliferation index (**B**). *** *p* < 0.001.

**Figure 5 ijms-23-02962-f005:**
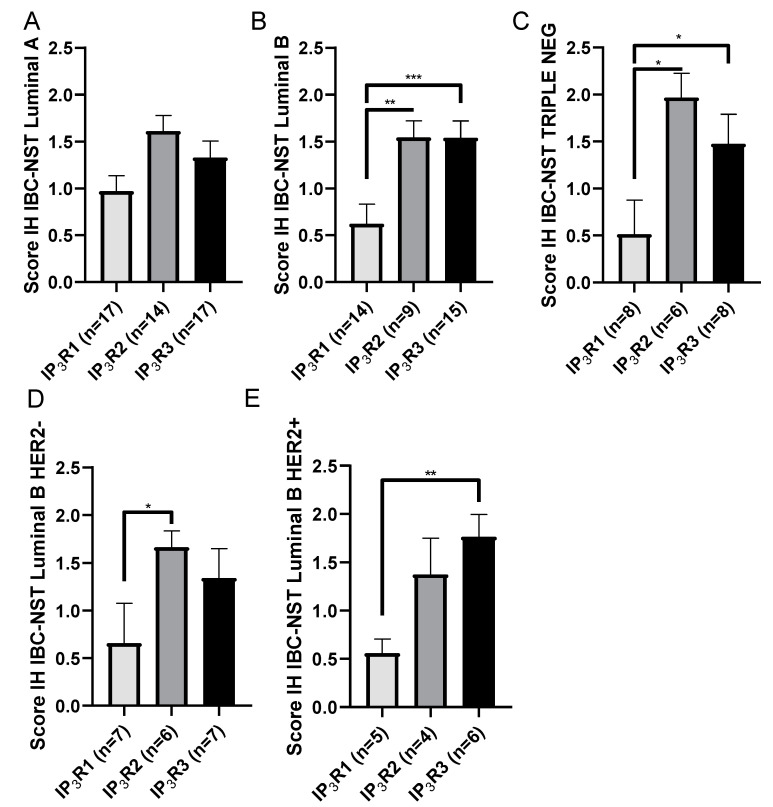
IP_3_R expression score as a function of the BC molecular subtype. (**A**) Luminal A, (**B**) luminal B, (**C**) triple negative, (**D**) luminal B HER2-, and (**E**) luminal B HER2+ BC. * *p* < 0.05; ** *p* < 0.01; *** *p* < 0.001.

**Figure 6 ijms-23-02962-f006:**
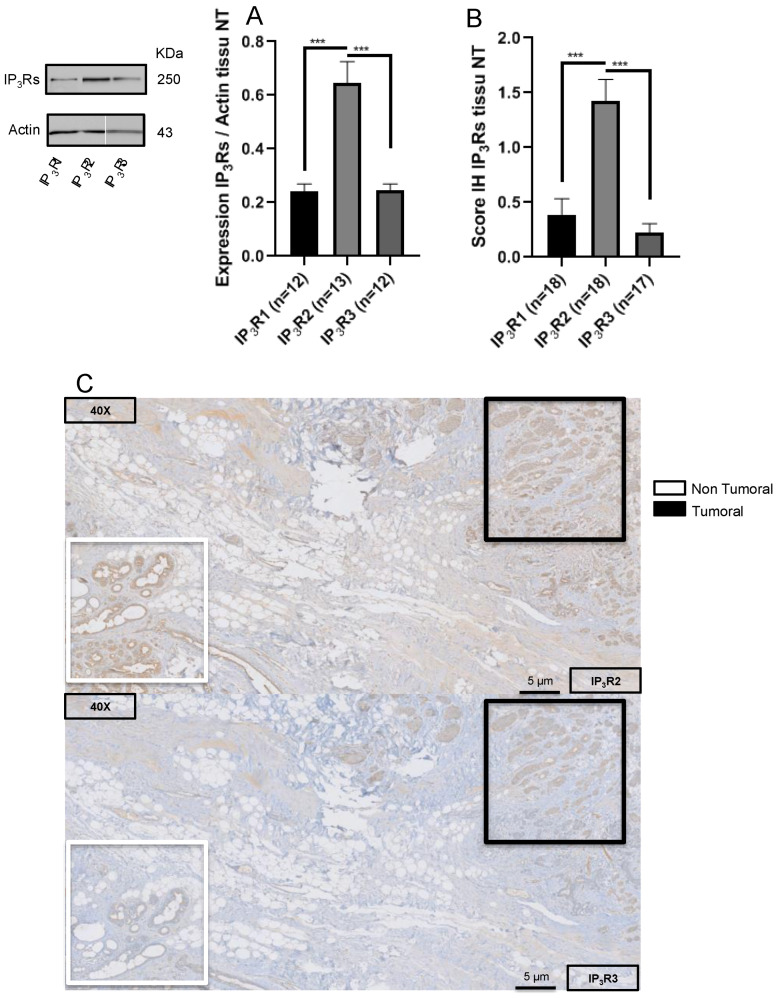
IP_3_R expression in non-tumor tissue. (**A**) Western blot analysis of the relative expression of IP_3_R1/2/3. (**B**) The IH expression score for IP_3_R1/2/3. (**C**) Representative IH images. *** *p* < 0.001.

**Figure 7 ijms-23-02962-f007:**
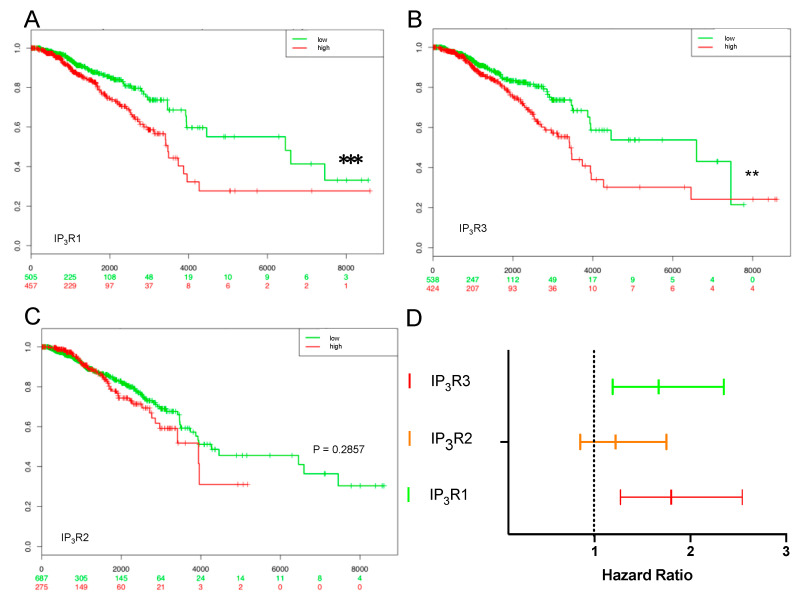
Analysis of the association between IP_3_R expression and overall survival, after application of the SurvExpress tool to the “*BRCA-TCGA Breast invasive carcinoma*—*July 2016*” database (comprising 962 BC samples). (**A**,**B**) IP_3_R1 expression and IP_3_R3 expression were significantly associated with poor overall survival (*p* = 0.0009 and *p* = 0.003). (**C**) IP_3_R2 expression was not significantly associated with poor overall survival (*p* = 0.286). (**D**) The HR for each IP_3_R appears in an analysis of the “*Breast invasive carcinoma*—*July 2016”* database. ** *p* < 0.01; *** *p* < 0.001.

**Table 1 ijms-23-02962-t001:** Breast cancer molecular subtype. ER: estrogen receptor, PR: progesterone receptor.

	ER	PR	HER2	Ki67 Index
**Luminal A**	+	+	−	Low
**Luminal B Her2−**	+	+	−	High
**Luminal B Her2+**	+	+	+	High
**Her2**	−	−	+	High
**Triple negative**	−	−	−	High

**Table 2 ijms-23-02962-t002:** Characteristics of the cohort of Invasive breast carcinoma of no special type (IBC-NST) samples from the CARCINO study and the Jean Godinot Institute cancer center. Data are quoted as the mean ± SD or n (%). BMI: body mass index, T: tumor size (T1: ≤20 mm; T2: 20–50 mm; T3: 50 mm). N: regional lymph nodes (N0: no lymph node invasion, N+: lymph nodes invaded). HmR: hormone receptor. SBR: Scarff-Bloom-Richardson.

		CARCINO IBC-NST Samples (n = 52)	Jean Godinot Institute IBC-NST Samples (n = 15)
		n (%)	n (%)
Age		57 ± 1.7	65.3 ± 3
BMI		27.1 ± 0.75	28.7 ± 2.4
	T1	16 (30.7)	4 (26.7)
	T2	31 (59.6)	8 (53.3)
TNM	T3	5 (9.7)	3 (20)
	N0	25 (48.1)	8 (53.3)
	N+	27 (51.9)	7 (46.7)
HmR+		43 (82.7)	12 (80)
HER2+++		12 (23.1)	2 (13.3)
Triple-negative		7 (13.5)	2 (13.3)
	1	6 (11.5)	2 (13.3)
SBR grade	2	26 (50)	3 (20)
	3	20 (38.5)	10 (66.7)
Ki67 > 20%		26 (50)	10 (66.7)

**Table 3 ijms-23-02962-t003:** Statistically significant correlations between IP_3_R subtype expression and survival. IP_3_R3 and IP_3_R1 expression levels were significantly associated with worse overall survival in three datasets (comprising 1358 patients (A) and 1700 patients (B), respectively), and IP_3_R2 was significantly associated with worse overall survival in two datasets (comprising 434 patients) (C). Expression of the three IP_3_R subtypes was significantly associated with worse recurrence-free survival in two datasets (comprising 3449 patients) (D). HR: hazard ratio. * *p* < 0.05; ** *p* < 0.01; *** *p* < 0.001.

**A**			
**Datasets—IP_3_R3 and Overall Survival**	**N: Low-Risk Group vs. High-Risk Group**	**HR [95%CI]**	***p*-Value**
Breast—Breast cancer recurrence data, 9 datasets from 7 authors	198; 92 vs. 106	1.74 [1.23–2.46]	******
Breast—Breast Cancer Metabase:10 cohorts 22K genes	198; 86 vs. 112	1.44 [1.02–2.02]	*****
Breast—BRCA-TCGA Breast Invasive Carcinoma—July 2016	962; 538 vs. 424	1.67 [1.19–2.35]	******
**B**			
**Datasets—** **IP_3_R1 and Overall Survival**	**N: Low-Risk Group vs. High-Risk Group**	**HR [95%IC]**	** *p* ** **-Value**
Breast—Breast Invasive Carcinoma TCGA	502; 376 vs. 126	1.87 [1–3.19]	*****
Breast—Miller Bergh Breast GSE3494−GPL96	236; 45 vs. 191	4.64 [1.45–14.87]	******
Breast—BRCA−TCGA Breast invasive carcinoma—July 2016	962; 505 vs. 457	1.8 [1.27–2.54]	*******
**C**			
**Datasets—** **IP_3_R2 and Overall Survival**	**N: Low-Risk Group vs. High-Risk Group**	**HR [95%IC]**	** *p* ** **-Value**
Breast—Breast Cancer Metabase:10 cohorts 22K genes	198; 167 vs. 31	1.61 [1.03–2.53]	*****
Breast—Miller Bergh Breast GSE3494−GPL96	236; 45 vs. 191	4.64 [1.45–14.87]	******
**D**			
**Datasets—** **Disease-Free Survival**	**N: Low-Risk Group vs. High-Risk Group**	**HR [95%IC]**	** *p* ** **-Value**
IP_3_R3—Breast cancer recurrence data, 9 datasets from 7 authors	1561; 967 vs. 594	1.28 [1.08–1.51]	******
IP_3_R3—Breast Cancer Metabase:10 cohorts 22K genes	1888; 1407 vs. 481	1.25 [1.05–1.49]	*****
IP_3_R2—Breast cancer recurrence data, 9 datasets from 7 authors	1561; 1194 vs. 367	1.27 [1.05–1.53]	*****
IP_3_R2—Breast Cancer Metabase:10 cohorts 22K genes	1888; 1614 vs. 274	1.36 [1.11–1.67]	******
IP_3_R1—Breast cancer recurrence data, 9 datasets from 7 authors	1561; 842 vs. 719	1.43 [1.21–1.69]	*******
IP_3_R1—Breast Cancer Metabase:10 cohorts 22K genes	1888; 1113 vs. 775	1.48 [1.27–1.73]	*******

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
