# Peer review of "Inositol (1,4,5)-Trisphosphate Receptors in Invasive Breast Cancer: A New Prognostic Tool?"

_ijms, 2022, doi:10.3390/ijms23062962_

Round 1
Reviewer 1 Report
The present manuscript analyzes the potential value of the three IP3R subtypes as prognostic biomarkers in breast cancer, together with a complementary analysis of public databases to investigate the survival and progression-free survival rates in patients. As a conclusion, they confirm that mainly IP3R3 expression is upregulated in this type of cancer and that IP3R3 is the only subtype correlated with prognostic factors and the cancer aggressiveness. In my opinion, the study looks interesting for its future use in clinical applications. There are however, few clarifications and modification required before further consideration.
Questions and concerns to be addressed properly:
- Introduction:
- The introduction chapter begins with some data about breast cancer in France, which is not referenced, maybe it is interesting to add the survey used to obtain that information. Furthermore, the data is from 2018. Being currently in 2022, more updated information, if any, would be appreciated.
- Sometimes percentages are written with symbol and other times in letter, why not to homogenize the style? Example, line 117 “ninety percent” and line 129 “5% or 15%”.
-In the last two paragraphs of the introduction, authors start (line 158) describing IP3Rs expression in different oncogenic processes, and then in line 172 they explain “with regard to BC more specifically”. The thing is that in the previous paragraph they also talk about breast cancer, so maybe it makes more sense to separate properly both paragraphs grouping on one side information of other oncogenic processes and on the other side breast cancer.
-During the introduction, the authors only refer to women breast cancer. Perhaps it would be important to consider adding data on male breast cancer. Even if its incidence is infinitely lower comparing with female breast cancer, it is also a health problem that is on the rise and therefore should also be considered.
- Materials and methods:
- On the first paragraph, the authors explain how the tumor tissue samples were collected. In line 185, they say, “After resection, the pathologist collected one to three tumor tissue samples. At least one sample per patient included was frozen immediately and stored at -80ºC”. Is this freezing step required for something special? What happens with the other 2 samples collected? Were they processed for western blot or other experimentation immediately?
-Revise used materials, as sometimes they have no reference and brand of the company.
-Used secondary antibodies for western blot are not specified, neither the used concentration nor dilution (line 207).
-For the immunohistochemistry and specifically the part that authors explain the molecular subtype assessment (from line 236 to line 243), maybe it is clearer to specify the + and – characteristics in a table, were all that information is summarized and could be easier to visualize, the differences between subtypes.
-When describing the databases used for the survival analysis (line 245), the database link (if exists) could be included and the last visited date, as currently the included patients number could be increased.
- Results:
-On table 1 (line 279), the description of the used acronyms, would be easier to follow if they are in the order they appear (BMI, TNM, T1, T2, T3, N0, N+, HmR+, HER2+++, SBR, Ki67). Furthermore, some explained acronyms do not appear in the table (N1, N2, N3, M1).
-On Figure 1, representative western blot images need to be add together with the histogram to facilitate the observation of the result. Furthermore, no data of expression values and IH values are added in the text, neither the p values.
- Maybe I am wrong but the described significant changes in Figure 1C are difficult to appreciate visually. It seems that IP3R2 and IP3R3 are increased in tumoral tissue compared with non tumoral, instead of IP3R1, which seems practically same expression. Change the images to better show what the authors describe as result.
-In figure 2, 3, 4 and 5, the comparison values scores need to be explain on the text and not in the figure foot. Moreover decide the use of *** for significance or the use of numbers. Usually *** is included in the figure and (* p<0,05; ** p<0,01; *** p<0,001) is establish on the figure foot. I do not understand why in figure 3 the value for p=0,08 is added to the figure and commented as important result, when it is not significant.
- On Figure 6, representative western blot images need to be add together with the histogram to facilitate the observation of the result. Furthermore, no data of expression values and IH values are added in the text, neither the p values. Do the authors have any other image were the explained differences are observed more appropriately?
- Discussion:
- Following the comments on the introduction, I expect that all the results obtained in the study to be from women breast cancer tumors. These results could be extrapolated to male breast cancer? In my opinion, the discussion should be extended to discuss this proposed topic.
Author Response
We are grateful to the reviewer for giving us the chance to improve our paper. We found the comments very interesting and very helpful.
- Introduction:
The introduction chapter begins with some data about breast cancer in France, which is not referenced, maybe it is interesting to add the survey used to obtain that information. Furthermore, the data is from 2018. Being currently in 2022, more updated information, if any, would be appreciated.
We thank the reviewer for this very relevant comment. We added the most recent data for breast cancer epidemiology worldwide and removed the French data.
Sometimes percentages are written with symbol and other times in letter, why not to homogenize the style? Example, line 117 “ninety percent” and line 129 “5% or 15%”.
We apologize for this heterogeneity. The style of the percentages have been homogenized.
In the last two paragraphs of the introduction, authors start (line 158) describing IP3Rs expression in different oncogenic processes, and then in line 172 they explain “with regard to BC more specifically”. The thing is that in the previous paragraph they also talk about breast cancer, so maybe it makes more sense to separate properly both paragraphs grouping on one side information of other oncogenic processes and on the other side breast cancer.
Changes have been made.
-During the introduction, the authors only refer to women breast cancer. Perhaps it would be important to consider adding data on male breast cancer. Even if its incidence is infinitely lower comparing with female breast cancer, it is also a health problem that is on the rise and therefore should also be considered.
This is a very interesting remark because the existence of breast cancer in men is often overlooked. Nevertheless, we purposely excluded male breast cancers from our introduction due to the lack of samples for our study for which we only included breast cancer samples from female patients. Male breast cancer is extremely rare and is characterized by a worse prognosis. Therefore, this could constitute a bias.
Moreover, because of its rarity, male breast cancer samples were not available in TCGA datasets or in our tissue biobank. We deplore this lack of data but have chosen to focus our comments on breast cancer in women for which we have sufficient statistically usable data.
- Materials and methods:
- On the first paragraph, the authors explain how the tumor tissue samples were collected. In line 185, they say, “After resection, the pathologist collected one to three tumor tissue samples. At least one sample per patient included was frozen immediately and stored at -80ºC”. Is this freezing step required for something special? What happens with the other 2 samples collected? Were they processed for western blot or other experimentation immediately?
We used one of the samples for primary culture. Unfortunately we have not been able to collect usable data in number and quality for the moment. The frozen samples allowed us to do the WB experiments. The third sample was used for immunohistochemistry.
-Revise used materials, as sometimes they have no reference and brand of the company.
We apologize for this lack. Modifications have been made.
-Used secondary antibodies for western blot are not specified, neither the used concentration nor dilution (line 207).
Once again, we apologize for this lack. Modifications have been made.
-For the immunohistochemistry and specifically the part that authors explain the molecular subtype assessment (from line 236 to line 243), maybe it is clearer to specify the + and – characteristics in a table, were all that information is summarized and could be easier to visualize, the differences between subtypes.
A table has been made according to this relevant suggestion (Table 1, line 249).
-When describing the databases used for the survival analysis (line 245), the database link (if exists) could be included and the last visited date, as currently the included patients number could be increased.
We would have liked to be able to attach such information, unfortunately the use of survexpress is no longer available. Therefore we cannot provide a valid link here.
- Results:
-On table 1 (line 279), the description of the used acronyms, would be easier to follow if they are in the order they appear (BMI, TNM, T1, T2, T3, N0, N+, HmR+, HER2+++, SBR, Ki67). Furthermore, some explained acronyms do not appear in the table (N1, N2, N3, M1).
We thank the reviewer for this remark. Modifications have been made.
-On Figure 1, representative western blot images need to be add together with the histogram to facilitate the observation of the result.
Western blot images have been added. We hope the figure is now more readable
Furthermore, no data of expression values and IH values are added in the text, neither the p values.
In the text corresponding to figure 1 the expression and score values are present. A p value was missing and has been added according to the reviewer’s suggestion.
- Maybe I am wrong but the described significant changes in Figure 1C are difficult to appreciate visually. It seems that IP3R2 and IP3R3 are increased in tumoral tissue compared with non tumoral, instead of IP3R1, which seems practically same expression. Change the images to better show what the authors describe as result.
We apologize for this given back. This feeling is explained by the staining of the myoepithelial cells within the vessels and ducts. When looking only at healthy epithelial cells versus cancer cells the difference appears more clearly to our point of view. We accentuated the contrast of the images and hope it will make it more visible. Nevertheless, this analysis was validated by pathologist from CHU Amiens Picardie (Dr Pierre Rybarczyk and Pr Henri Sevestre)
-In figure 2, 3, 4 and 5, the comparison values scores need to be explain on the text and not in the figure foot. Moreover decide the use of *** for significance or the use of numbers. Usually *** is included in the figure and (* p<0,05; ** p<0,01; *** p<0,001) is establish on the figure foot.
We apologize for this confusion. Modifications have been made to standardize the representation.
I do not understand why in figure 3 the value for p=0,08 is added to the figure and commented as important result, when it is not significant.
We chose to put (p=0.08) because we think this result was interesting even though we only observed a trend that almost reached statistical significance. It is plausible that with a slightly larger number of patients we would have had a significant result.
- On Figure 6, representative western blot images need to be add together with the histogram to facilitate the observation of the result.
We completely agree. The images have been added.
Furthermore, no data of expression values and IH values are added in the text, neither the p values.
We apologize for this lack. Data have been added.
Do the authors have any other image were the explained differences are observed more appropriately?
We apologize for this appreciation. All the pictures taken show the same pattern. We re-opened our images in appropriate software and accentuate the contrast which ameliorate the visibility to our point of view. We hope it will help to the readability of this figure.
- Discussion:
Following the comments on the introduction, I expect that all the results obtained in the study to be from women breast cancer tumors. These results could be extrapolated to male breast cancer? In my opinion, the discussion should be extended to discuss this proposed topic.
We thank the reviewer for this interesting comment. As the reviewer may know,breast cancer in men remains extremely rare and the prognosis is different from that in women. Male breast cancers are generally more aggressive, which could constitute a bias in this type of study with survival curves analyses. For now, with the little data we have, we cannot unfortunately know if these results could be extrapolated to men. We did not have any cancer extracts from male patients in the tissue biobank until know, but we hope such comparison could be included in future studies.
Reviewer 2 Report
It is an excellent paper, with a good rational, methodology results with possible clinical impact.
Author Response
We thank the reviewer for the encouraging comments. We have also made some changes based on the comments of other reviewers, which we hope will improve our study.
Round 2
Reviewer 1 Report
Thanks to the authors for taking the suggestions into account. This new version is much better understood and I consider it suitable for publication with minor revision.
- I think it is appropiate to include the authors clarification about the freezing step and the use of the 3 samples different samples they take for each patient, as they do in the authors response report.
- In respect to the added WB images, in figure 1 there are some mistakes in the name and size of the samples and bands. There are some question marks, maybe due to pdf conversion problems. It is important to revise that before publication. In the WB added in figure 6 it seems there is not that kind of error.
Author Response
We thank the reviewer for this remark and are grateful for the review which profoundly improves our study.
I think it is appropriate to include the authors clarification about the freezing step and the use of the 3 samples different samples they take for each patient, as they do in the authors response report.
We hadn't thought of adding it to revision 1 and have now integrated it into the materials and methods section. This clarification brings more clarity to the understanding of our protocol.
In respect to the added WB images, in figure 1 there are some mistakes in the name and size of the samples and bands. There are some question marks, maybe due to pdf conversion problems. It is important to revise that before publication. In the WB added in figure 6 it seems there is not that kind of error.
We thank the reviewer for his attention and we deeply apologize for this error due to insufficient quality of images when converting to pdf. The problem is now fixed and the figures appear correctly.